# Persistence of SARS-CoV-2 Viral RNA in Nasopharyngeal Swabs after Death: An Observational Study

**DOI:** 10.3390/microorganisms9040800

**Published:** 2021-04-10

**Authors:** Francesca Servadei, Silvestro Mauriello, Manuel Scimeca, Bartolo Caggiano, Marco Ciotti, Lucia Anemona, Manuela Montanaro, Erica Giacobbi, Michele Treglia, Sergio Bernardini, Luigi Tonino Marsella, Nicoletta Urbano, Orazio Schillaci, Alessandro Mauriello

**Affiliations:** 1Anatomic Pathology, Department of Experimental Medicine, University of Rome “Tor Vergata”, Via Montpellier 1, 00133 Rome, Italy; francescaservadei@gmail.com (F.S.); anemona@uniroma2.it (L.A.); manuela.montanaro@uniroma2.it (M.M.); ericagiacobbi@gmail.com (E.G.); alessandro.mauriello@uniroma2.it (A.M.); 2Forensic Medicine, Department of Biomedicine and Prevention, University of Rome “Tor Vergata”, Via Montpellier 1, 00133 Rome, Italy; mauriel@uniroma2.it (S.M.); bartolocaggiano@gmail.com (B.C.); michelemario@hotmail.it (M.T.); marsella@uniroma2.it (L.T.M.); 3Nuclear Medicine, Department of Experimental Medicine, University of Rome “Tor Vergata”, 00133 Rome, Italy; orazio.schillaci@uniroma2.it; 4Saint Camillus International University of Health Sciences, Via di Sant’Alessandro, 8, 00131 Rome, Italy; 5Division of Clinical Biochemistry and Clinical Molecular Biology, Department of Experimental Medicine, University of Rome “Tor Vergata”, 00133 Rome, Italy; marco.ciotti@ptvonline.it (M.C.); bernardini@med.uniroma2.it (S.B.); 6Nuclear Medicine, Department of Biomedicine and Prevention, University of Rome “Tor Vergata”, 00133 Rome, Italy; n.urbano@virgilio.it; 7Tor Vergata Oncoscience Research (TOR), University of Rome “Tor Vergata”, 00133 Rome, Italy

**Keywords:** SARS-CoV-2, pandemic, autopsy, medico-legal procedures

## Abstract

The aim of this study was to investigate the persistence of SARS-CoV-2 in post-mortem swabs of subjects who died from SARS-CoV-2 infection. The presence of the virus was evaluated post-mortem from airways of 27 SARS-CoV-2 positive patients at three different time points (T1 2 h; T2 12 h; T3 24 h) by real-time PCR. Detection of antibodies to SARS-CoV-2 was performed by Maglumi 2019-nCoV IgM/IgG chemiluminescence assay. SARS-CoV-2 viral RNA was still detectable in 70.3% of cases within 2 h after death and in 66,6% of cases up to 24 h after death. Our data showed an increase of the viral load in 78,6% of positive individuals 24 h post-mortem (T3) in comparison to that evaluated 2 h after death (T1). Noteworthy, we detected a positive T3 post-mortem swab (24 h after death) from 4 subjects who were negative at T1 (2 h after death). The results of our study may have an important value in the management of deceased subjects not only with a suspected or confirmed diagnosis of SARS-CoV-2, but also for unspecified causes and in the absence of clinical documentation or medical assistance.

## 1. Introduction

The severe acute respiratory syndrome coronavirus-2 (SARS-CoV-2) is a novel coronavirus detected in December 2019 as the causative agent of a human respiratory infection (coronavirus disease 2019 or SARS-COV-2) in Wuhan, China [1,2,3]. Following the spreading of the infection worldwide, The World Health Organization (WHO) on 11 March 2020 declared the SARS-CoV-2 outbreak a global pandemic [4]. According to WHO reports [5], more than 133 million of global cases have been confirmed, with 2.90 million of deaths. Most recent epidemiological data showed that COVID-19 infection cause mild symptoms or no symptoms in at least 80% of patients [6]. In line with this evidence, several COVID-19 positive subjects were not identified from the tracking network [6]. It is conceivable that several subjects with a pre-existing asymptomatic COVID-19 infection underwent autopsy for sudden or unexplained death thus determining a great biological risk for workers involved in the autoptic exam. The most common transmission route for COVID-19 infection is person-to-person, mainly via respiratory aerosols and / or droplets [7]. Virus-contaminated surfaces can also be a potential source of infection [8]. However, it is difficult to establish the level of infectivity of each positive subject [7]. Indeed, despite available evidence from national contact tracing suggests that asymptomatically infected subjects are much less likely to transmit the virus than those who show symptoms, numerous cases of infections from asymptomatic subjects have been described [9]. While several guidelines have been issued regarding autopsy protocols in cases of confirmed or suspected COVID-19 deaths [10,11], there are limited recommendations regarding scene investigative protocols and morgue biosafety practices [6]. Moreover, to the best of our knowledge, only few studies investigated the persistence of SARS-CoV-2 infectious particles in the airways of decedents [7,11]. In forensic and pathology scenario, as in case of any other infectious disease, it is crucial to establish whether SARS-CoV-2 is capable to replicate after death of the infected individuals and, thereby, potentially transmit the infection [12]. These data would be essential in order to a) guarantee the safety protection for the examination team, b) counteract the progression of SARS-Cov-2 pandemic and c) improve the knowledge about the molecular mechanisms of SARS-Cov-2 infection. 

Starting from these considerations, the aim of this study was to investigate the persistence of SARS-CoV-2 in post-mortem swabs of subjects who died from COVID-19 infection.

To this end, the presence of the virus was evaluated post-mortem in airways of subjects who died from COVID-19 infection at three different time points (2 h–12 h– 24 h after death).

## 2. Materials and Methods

From 17 April 2020 to 25 June 2020, 27 patients (12 females, 15 males, mean age 76.2 +/-13.7) who died from COVID-19 pneumonia were referred to our Institution. Consecutive nasopharyngeal swabs were obtained from these subjects according to the following study design: swab within 2 h of death (T1), swab 12 h after death (T2) and third swab 24 h after death (T3). Also, blood samples were collected simultaneously with the first swab (T1) to evaluate the presence of SARS-CoV-2 IgG and IgM antibodies. The following laboratory parameters were evaluated with reference to the last blood sample taken and analysed ante mortem: (white blood cells (WBC) (g/dL), neutrophils (103 /μL), haemoglobin (Hb) (g/dL), lymphocytes (103 /μL), C Reactive Protein (CRP) (mg/dL), procalcitonin (PCT) (ng/mL), lactate dehydrogenase (LDH) (U/L) and pH). Data of the comorbidities, as well as the duration of their hospitalization were obtained from the examination of the medical records of all 27 patients.

### 2.1. Detection of SARS-CoV-2 RNA by Real-Time PCR

The presence of SARS-CoV-2 RNA was tested on nasopharyngeal swabs by real-time reverse transcriptase (qRT-PCR). using the AllplexTM 2019n-CoV assay (Seegene, Seoul, South Korea). This test was designed for the qualitative detection of the novel coronavirus in respiratory samples. RNA extraction and qRT-PCR set-up were carried out on NIMBUS (Seegene), an automated liquid handling workstation. Real-time qRT-PCR was performed on a CFX96TMDx platform (Bio-Rad Laboratories, Inc.,Irvine, CA, USA) followed by interpretation by Seegene’s Viewer Software. The Allplex™ 2019n-CoV assay is a multiplex qRT-PCR targeting the common envelope (E) gene, the specific nucleocapsid (N) and RNA-dependent-RNA-polymerase (RdRp) genes complying with the international validated protocols.

According to latest WHO recommendations [13], screening by qRT-PCR of a single discriminatory target was considered sufficient. Therefore, if one target tested positive, the case would be considered laboratory-confirmed [13].

The results of qRT-PCR test are showed as cycle-threshold (Ct) values. The Ct is defined as the number of cycles of amplification required for the fluorescent signal to cross the threshold, which is above the background signal (a low-level signal that is present in the assay regardless of whether target is present). Therefore, Ct values are inversely proportional to the amount of target nuclei acid present in the tested sample. Recent studies demonstrated that qRT-PCR Ct values correlate strongly with cultivable virus, providing a valuable surrogate for infectious virus detection in biological samples [14]. Accordingly, we defined a Ct-value less than 37 as a positive test, while a result of more than 40 was considered negative. Values between 37 and 40 cycles were considered equivocal, requiring confirmation by retesting. The obtained Ct values have been then grouped in 3 categories, defined as follows: <25 strongly positive; 25–35 moderately positive; >35 weakly positive [15].

### 2.2. Detection of SARS-CoV-2 Antibodies

Detection of antibodies to SARS-CoV-2 was performed by Maglumi 2019-nCoV IgM/IgG chemiluminescence assay, which is designed for the qualitative detection of IgM/IgG to SARS-CoV-2 in serum samples (SNIBE Diagnostics, Shenzhen, China). The assay is a capture chemiluminescence immunoassay for IgM and an indirect chemiluminescence immunoassay for IgG. The test uses magnetic microbeads coated with 2019-nCoV recombinant antigen (nucleocapsid protein, NP and Spike Protein, SP) and anti-human IgG and IgM antibodies labeled with N-(aminobutyl)-N-(ethylisoluminol) (ABEI) that form complexes. The light signal measured by a photomultiplier as relative light units (RLUs), is proportional to the concentration of 2019-nCoV IgG and IgM present in the sample. A result < 1.00 AU/mL is considered negative, whereas the sample is defined reactive if result ≥ 1.00 AU/mL.

### 2.3. Statistical Analysis

Data were analyzed using SPSS version 21.0 (SPSS Inc, Chicago, IL, USA) software. Continuous variables were expressed as the mean ± SD. The Shapiro-Wilk test was used to statistically assess the normal distribution of the data. Comparisons between continuous variables were performed using the independent Student *t*-test or the Wilcoxon rank sum test. Categorical data were analyzed using the chi square test or the Fisher exact test.

A linear regression was performed in order to evaluate the possible correlation between the duration of hospitalization and value of IgM and IgG.

A 2-tailed *p* value < 0.05 was considered statistically significant.

## 3. Results

### 3.1. Clinical Data.

27 patients with COVID-19 interstitial pneumonia were included in this study, 15 males and 12 females (age range 44–95, mean 76.2 +/- 13.7). The duration of hospitalization ranged from 1 to 112 days (mean 23.7 +/- 25.7). All patients had more than one associated disease (13 had heart failure, arrhythmia or ischemic heart disease, 11 hypertension, 11 dementia, 7 chronic renal failure, 6 neoplasms, 5 diabetes, 2 obesity, 1 recent kidney transplantation, 1 stroke) (Table 1).

### 3.2. Analysis of Samples.

Data from last ante-mortem swabs were available for all 27 patients and in all cases were positive for at least one target gene. In particular, all patients tested positive for N gene (CT values ranged from 13 to 38), 24 of them tested positive also for R gene (CT values from 12 to 35) and 22 of them were positive for E gene (CT values from 9 to 30) (Table 1).

A total of 81 swabs were prospectively collected post-mortem from the nasopharynx of the 27 patients. For each patient 3 post-mortem swabs were collected: within 2 h after death (T1), after 12 h (T2) and after 24 h (T3). Of the 27 patients with the last positive ante-mortem swab, 19 (70.3%) were still positive at the first post-mortem swab (T1), while 8 cases (29.7%) tested negative at T1. In particular, 12 were positive for the E gene (Ct values < 25 in 8 cases, between 25 and 35 in 4 cases), 16 for the R gene (Ct values < 25 in 8 cases, 25–35 in 7 cases and > 35 in 1 case) and 18 for the N gene (Ct values < 25 in 7 cases, 25–35 in 10 cases and > 35 in 1 case) (Table 1).

In patients who remained positive at T1, an inverse correlation with length of hospitalization was observed. In fact, in this group the mean duration of hospitalization was 13 days (+/- 14) while in patients with negative T1 swab the mean duration was 47 days (+/- 32; *p*: 0.002). There was no statistically significant correlation between the length of hospitalization and the IgM and IgG serum values (R2 −0.03, *p* = 0.51; R2 −0.04, *p* = 0.58, respectively). Four cases showed a negative result for serum IgM and IgG, even though they were repeatedly positive for nasopharyngeal swabs. Although a false negative cannot be excluded, the data can also be interpreted in light of the immunosuppressive condition reported in these four patients.

Among the different clinical and laboratory investigated parameters, we observed a significant correlation between lower Ct values at T1, indicative of higher levels of viral load, and both higher PCT and LDH values detected in the last ante-mortem blood sample. [1.23 +/- 0.7 (in patients with Ct < 25) vs 0.74 +/- 0.15 (in patients with Ct > 25) and 662.80 +/- 146 (in patients with Ct < 25) vs 349.37 +/- 54.98 (in patients with Ct > 25), respectively for PCT and LDH]. No further significant correlations were found. Also, no significant association was observed by comparing the Ct values of E, N and R genes at both T1 and T3 and some inflammatory biomarkers such as PCR lymphocytes and neutrophils (see table in Appendix A).

Considering the swabs performed 12 h after death (T2), 16 patients tested positive for at least 1 gene (13 for the E gene, 16 for the R gene and 16 for the N gene). Noteworthy, among these 16 patients, 2 resulted negative at T1 swab, while positive at last ante-mortem swab.

Considering the nasopharyngeal swabs performed 24 h after death (T3), 18 patients tested positive for at least 1 gene (17 for the E gene, 17 for the R gene and 18 for the N gene). Among these 18 patients, 4 resulted negatives at T1 swab, while positive at last ante-mortem swab.

In addition, among the 19 positive patients at T1, 5 tested negatives at T3 and 14 remain positive. Of the 14 patients who remained positive at T3, 11 (78.6%) showed a reduction in Ct values and therefore an increase in viral load after 24 h. Out of the 8 cases who were negative at T1, 4 resulted positive at T3 (Ct values at T3: 20 in 1 case, 30 in 2 cases and 37 in 1 case).

## 4. Discussion

In this study, 3 subsequent post-mortem nasopharyngeal swabs in 27 patients who died for confirmed SARS-COV-2 pneumonia were performed to investigate the possible persistence of SARS-CoV-2 viral RNA in the upper respiratory tract. Amazingly, SARS-CoV-2 viral RNA was still detectable in 70.3% of cases within 2 h after death and in 66.6% of cases up to 24 h after death. A previous study describes the persistence of viral RNA in pharyngeal samples after death in a lower number of cases, reporting a slow decrease in viral load 24 h after death and, in one case, the presence of the virus at 128 h after death [7]. The results here reported confirm these data but, interestingly, no decrease in positivity up to 24 h after death was found. On the contrary, our data showed an increase of the viral load in 78.6% of positive individuals 24 h post-mortem (T3), in comparison to that evaluated 2 h after death (T1). Noteworthy, we detected a positive T3 (24 h after death) post-mortem swab from 4 subjects who were negative at T1 (2 h after death). There are some possible interpretations about this funding: the possibility of a false negative for the first test should be considered. Nowadays it is known that inappropriate sample taking or transport, as well as a low viral load below the sensitivity limit, may affect the results [16]. In order to control the sampling procedures, a single operator was trained and designated for the execution of all nasopharyngeal swabs [17]. Nevertheless, also post-mortem transformative phenomena cannot be excluded among all the possible explanations. Indeed, the post-mortem cell lysis and / or the increase in chest pressure due to the formation of intestinal gas which cause the compression on the diaphragm, might explain the release of the alveolar contents and virus-containing secretions in the upper airways. Also, in absence of circulating immune cells, during a post-mortem time window, it is conceivable that some spike positive human cells can be infected by SARS-COV-2 causing the release of a great amount of viral particles. According to literature, several studies on animal models reported a resilience and resistance of some cells to remain alive, also increasing their activity even after death [18]. These results are effectively in line with the “Twilight of Death”, the now accepted concept which explain the existence of a time window between death and body’s decomposition, where not all the body’s cells are yet dead. In concert with this theory, many authors suggest considering the death phenomenon as a slow shutdown process rather than a simple switch-off. Thus, pathological process such as viral infections could continue, or even increase, in the first hours after death.

The important results obtained in this study require to adopt appropriate protective measures by both forensic pathologists and morgue operators. To this end, when a diagnostic autopsy is required, one post-mortem swab is not sufficient to declare a subject negative for the presence of SARS-COV-2; at least two or three consecutive swabs are necessary to exclude false negatives, especially for those cases where clinical data are not available (i.e., patients died during the transportation at the hospital or without medical intervention).

From an epidemiological point of view, our study confirms the greater lethality of the COVID-19 disease in elderly patients (22 out of 27 were aged more than 70 years) and in individuals with more than one comorbidity (100% of our sample). Furthermore, an inverse correlation was found between the length of hospitalization and the positivity of T1 swab. In fact, among the 19 patients who were still positive at T1, the mean duration of hospitalization was 13 days, as compared with the 8 patients resulted negative at T1, which showed a mean of 47 days of hospitalization. This could be explained by the establishment of an abnormal systemic response leading to a more rapid fatal outcome in the first group of patients. On the contrary, the exitus of individuals with a longer duration of hospitalization can be traced back to the multi-organ impairment induced by the virus over a longer period, all aggravated by the simultaneous presence of serious comorbidities. No significant association has been found among the Ct values of E, N, R genes (T1) and some circulating biomarkers such as lymphocytes, neutrophils and PCR. As reported above, this could be explained by the influence of post-mortem phenomena on the viral load.

Another interesting result of our study is the association between low CT values in T1 and high ante-mortem PCT and lactate dehydrogenase LDH levels, that was statistically significant in our analysis. Procalcitonin is a precursor of the hormone calcitonin, which is involved in calcium homeostasis and is produced by proteolysis of pre-procalcitonin at the level of thyroid C cells and neuroendocrine cells of the lungs and intestines [19]. The level of serum procalcitonin in healthy individuals generally remains below the detection limit (10 pg/mL) of the clinical test, or it is in any case less than 0.05 ng/mL. As a consequence of the onset of serious bacterial infections and the related systemic inflammatory response patterns, procalcitonin is released intact by extra-thyroid organs, mainly by parenchymal cells and organs connected with the immune system, with a consequent increase in its blood values. Therefore, measurement of blood procalcitonin concentration can be used as a marker of severe bacterial infection and sepsis and well correlates with their severity. Lactate dehydrogenase is a cytoplasmatic enzyme presents in many tissues of the human body and is involved in the metabolism of glucose by catalyzing the conversion of pyruvic acid into lactic acid. In general, the finding of increased levels of LDH in the blood is evidence of tissue damage, such as tissue necrosis that can affect various organs, following different types of pathological processes [20]. Therefore, the association between high ante-mortem PCT and LDH values and low Ct values at T1 that emerged from our study represents the confirmation, respectively, of the severe septic state and of the important tissue and cellular damage, both related to a condition of multisystemic inflammation, typical of complicated COVID-19 infections.

## 5. Conclusions

The results of our study may have an important value in the management of deceased subjects not only for subjects with a suspected or confirmed diagnosis of COVID-19, but also for unspecified causes and in the absence of clinical documentation or medical assistance. In fact, the demonstration of SARS-CoV-2 positivity in the upper airways of decedents, also with a possible increase in viral load within 24 h of death, requires the use of adequate protective and preventive measures by healthcare professionals, forensic team and mortuary operators. In addition, the presence of potentially infectious viral particles in the upper airways of decedents 24h post-mortem, as well as the increase in the viral load observed in some subjects, could shed new light on the molecular mechanisms of COVID-19 infection.

## Figures and Tables

**Table 1 microorganisms-09-00800-t001:** Last ante-mortem and subsequent post-mortem nasopharyngeal swabs. Synthetic report of our cases, indicating age, gender, duration of hospitalization and categories of Ct values in last ante-mortem and subsequent post-mortem nasopharyngeal swabs, referred to as N for negative, L for low viral load of at least one gene (equivalent to Ct values > 35), M for moderate viral load of at least one gene (corresponding to Ct values between 25 and 35) and H for high viral load of at least one gene (Ct values < 25). The colors highlight the evolution of viral load in the three subsequent post-mortem swabs (unchanged, decreased and increased). In cases 18 and 27, even if the category remains moderately positive, there is a mild progressive increase in viral load.

Case	Age (yrs)	Gender	Duration of Hospitalization (Days)	Ante-Mortem Swab *	2 h Post- Mortem Swab *	12 h Post-Mortem Swab *	24 h Post-Mortem Swab *
1	95	M	9	H	L	N	N
2	87	F	1	H	H	N	N
3	78	F	17	M	M	N	N
4	73	F	59	H	H	N	N
5	86	M	73	H	N	N	N
6	72	M	19	H	M	M	H
7	75	M	20	H	H	H	H
8	82	M	8	H	H	H	H
9	55	M	31	H	H	H	H
10	86	M	1	H	H	H	H
11	85	M	42	H	N	N	N
12	44	F	22	H	L	M	H
13	89	F	2	M	N	M	M
14	82	M	13	L	N	N	L
15	76	M	7	H	H	H	H
16	47	M	18	H	N	H	H
17	57	M	8	H	M	M	H
18	87	F	5	H	M	M	M
19	75	M	112	H	N	N	M
20	86	M	10	M	M	M	M
21	72	F	27	M	N	N	N
22	55	M	4	H	H	N	H
23	85	F	27	H	M	M	H
24	91	F	51	L	N	H	M
25	83	M	41	H	N	N	N
26	78	F	14	M	M	M	N
27	75	F	23	H	M	M	M

* Legend: N = negative; L = > 35 Ct; M = 25–35 Ct; H = < 25 Ct. Grey Unchanged from 1^st^ to 3^rd^ sample; Orange Decreased viral load from 1st to 3rd sample; Yellow Increased from 1st to 3rd post-mortem sample.

## Data Availability

Data will be available on request.

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
