# Peer review of "Persistence of SARS-CoV-2 Viral RNA in Nasopharyngeal Swabs after Death: An Observational Study"

_microorganisms, 2021, doi:10.3390/microorganisms9040800_

Round 1
Reviewer 1 Report
In this original article entitled “Persistence of SARS-CoV-2 viral RNA in nasopharyngeal 2 swabs after death: an observational study” the authors aimed to assess the presence of SARS-CoV-2 viral load at the time of death and post-mortem 2 hours, 12 hours and 24 hours.
The aim of the article is of high interest, and it addresses the post-mortem SARS-Cov-2 diagnostics yield from nasopharyngeal swab samples.
I would have the following major points:
- The authors present in Table 1 the rate of SARS-CoV-2 positivity, but is this an aggregated data based on either N, R, or E gene detection?
- In the statistical analysis Part (LINES 119-128) the use of mean±SD or ±SEM should be specified, and the applied tests, should be mentioned in the manuscript where the results and correlations are presented.
- Did the patients have beside PCT and LDH other inflammatory markers detected (e.g. IL-6, TNF, ferritin, hsCRP, etc.) which are relevant in COVID-19? The correlation analysis between Ct values and these parameters would be of high relevance.
I have a few other minor remarks:
- In the whole manuscript the decimal point should be used for all values (e.g. 70.3% LINES 155, 132, 133, 156 etc.)
- I would recommend the use of “L” for liter (e.g. dL, nL, mL. e.g. LINES 76,77,78) in the manuscript according to the IUPAC recommendations.
- LINE 35: coronavirus disease 2019 should be abbreviated as COVID-19, the SARS-CoV-2 abbreviation should only be used, when referring to the virus itself.
- LINE 38: data should be updated with newer pandemic statistics.
- LINE 39: „SARS-CoV-2infection” a space is missing, correct to: „SARS-CoV-2 infection”
- LINE 40: „patients6” , 6 should be removed.
- LINES 83, 92, 95, 100: I would recommend the correct abbreviation for real-time reverse transcriptase PCR: qRT-PCR.
- LINE 83: please remove “.” after qRT-PCR abbreviation.

Author Response
Ref.: Manuscript: microorganisms-1175899
"Persistence of SARS-CoV-2 viral RNA in nasopharyngeal swabs after death: an observational study"
Submitted to: Microorganisms
Before we begin the point-by-point review of the list of concerns, we would like to thank the Reviewer for their comments on how to improve the manuscript, which has been revised accordingly, as well as the Editors for calling for a new submission of an improved version of our manuscript.
Reply to Reviewer 1
In this original article entitled “Persistence of SARS-CoV-2 viral RNA in nasopharyngeal 2 swabs after death: an observational study” the authors aimed to assess the presence of SARS-CoV-2 viral load at the time of death and post-mortem 2 hours, 12 hours and 24 hours.
The aim of the article is of high interest, and it addresses the post-mortem SARS-Cov-2 diagnostics yield from nasopharyngeal swab samples.
Reply: we would like to thank the Reviewer for expressing interest in our work.
I would have the following major points:
The authors present in Table 1 the rate of SARS-CoV-2 positivity, but is this an aggregated data based on either N, R, or E gene detection?
Reply: Thank you for this pointing out. In the new version of our manuscript, we better specified in the table legend that L indicates the positivity for at least one gene.
In the statistical analysis Part (LINES 119-128) the use of mean±SD or ±SEM should be specified, and the applied tests, should be mentioned in the manuscript where the results and correlations are presented.
Reply: Thank you for this suggestion. .In the new version of our manuscript, we modified the paragraph Statistical analysis according to reviewer suggestion also adding the test performed for comparing Ct values and serum inflammatory biomarkers.
Did the patients have beside PCT and LDH other inflammatory markers detected (e.g. IL-6, TNF, ferritin, hsCRP, etc.) which are relevant in COVID-19? The correlation analysis between Ct values and these parameters would be of high relevance.
Reply: Thank you for this pointing out. We re-evaluated the medical records of patients included in this study obtaining the serum values of some inflammatory biomarkers such as lymphocytes, neutrophils and PCR. We performed Pearson’s correlation to verify the possible association among the values of these inflammatory biomarkers and the Ct values of both the first and third swabs. However, no significant association was found. This could be explained by the influence of post-mortem phenomenon on the SARS-CoV-2 load as described in our study.
In the new version of our manuscript, we reported and discussed these data.
Results paragraph
Also, no significant association was observed by comparing the Ct values of E, N and R genes at both T1 and T3 and some inflammatory biomarkers such as PCR lymphocytes and neutrophils.
Discussion paragraph
No significant association has been found among the Ct values of E, N, R genes (T1) and some circulating biomarkers such as lymphocytes, neutrophils and PCR. As reported above, this could be explained by the influence of post-mortem phenomena on the viral load.
Only for the Reviewer we reported the table with the results of the analysis
Table 2 reports the R2 and p values of the Pearson’s correlation perform to evaluate the possible association among the Ct values of N, R and R genes at both T1 and T3 and the serum values of PCR, neutrophils and lymphocytes.
|
|
PCR |
Neutrophils |
Lymphocytes |
|
Swab within 2 hours of death (T1) |
|
|
|
|
N ct |
-0.04 (p=0.73) |
0.001 (p=032) |
-0.04 (p=0.99) |
|
Rd Rp Ct |
0.02 (p=0.25) |
-0.04 (p=0.85) |
-0.02 (p=0.46) |
|
E ct |
-0.05 (P=0.97) |
0.002 (p=0.32) |
0.02 (p=0.22) |
|
|
|
|
|
|
Swab 24 hours after death (T3) |
|
|
|
|
N ct |
-0.06 (p=0.69) |
0.09 (p=0.11) |
-0.06 (p=0.49) |
|
Rd Rp Ct |
0.06 (p=0.18) |
0.07 (p=0.14) |
-0.06 (p=0.96) |
|
E ct |
0.04 (p=0.22) |
0.09(p=0.13) |
-0.06 (p=0.98) |
I have a few other minor remarks:
- In the whole manuscript the decimal point should be used for all values (e.g. 70.3% LINES 155, 132, 133, 156 etc.)
- I would recommend the use of “L” for liter (e.g. dL, nL, mL. e.g. LINES 76,77,78) in the manuscript according to the IUPAC recommendations.
- LINE 35: coronavirus disease 2019 should be abbreviated as COVID-19, the SARS-CoV-2 abbreviation should only be used, when referring to the virus itself.
- LINE 38: data should be updated with newer pandemic statistics.
- LINE 39: „SARS-CoV-2infection” a space is missing, correct to: „SARS-CoV-2 infection”
- LINE 40: „patients6” , 6 should be removed.
- LINES 83, 92, 95, 100: I would recommend the correct abbreviation for real-time reverse transcriptase PCR: qRT-PCR.
- LINE 83: please remove “.” after qRT-PCR abbreviation.
Reply: done
Reviewer 2 Report
In this study was to investigate the persistence of SARS-CoV-2 in post-mortem 18 swabs of subjects who died from SARS-CoV-2 infection. The presence of the virus was evaluated 19 post-mortem from airways of 27 SARS-CoV-2 positive patients at three different time points ( 2 hours; 12 hours; 24 hours) by real-time PCR. The study is interesting but I want to ask if the authors did viral isolation to verify the presence of virus.
The authors have to correct line 40 and line 79
Author Response
Ref.: Manuscript: microorganisms-1175899
"Persistence of SARS-CoV-2 viral RNA in nasopharyngeal swabs after death: an observational study"
Submitted to: Microorganisms
Before we begin the point-by-point review of the list of concerns, we would like to thank the Reviewer for their comments on how to improve the manuscript, which has been revised accordingly, as well as the Editors for calling for a new submission of an improved version of our manuscript.
Reply to Reviewer 2
In this study was to investigate the persistence of SARS-CoV-2 in post-mortem 18 swabs of subjects who died from SARS-CoV-2 infection. The presence of the virus was evaluated 19 post-mortem from airways of 27 SARS-CoV-2 positive patients at three different time points (2 hours; 12 hours; 24 hours) by real-time PCR. The study is interesting but I want to ask if the authors did viral isolation to verify the presence of virus.
Reply: we would like to thank the Reviewer for expressing interest in our work. Unfortunately, the virus isolation was not included in our experimental design since the main interest of this study was to evaluate the persistence of viral particles after death according to common guidelines (evaluation of nasopharyngeal swabs by PCR).
The authors have to correct line 40 and line 79
Reply: done